# Enhancing Dataset Distillation with Concurrent Learning: Addressing Negative Correlations and Catastrophic Forgetting in Trajectory Matching

## Abstract

Dataset distillation generates a small synthetic dataset on which a model is trained to achieve performance comparable to that obtained on a complete dataset. Current state-of-the-art methods primarily focus on Trajectory Matching (TM), which optimizes the synthetic dataset by matching its training trajectory with that from the real dataset. Due to convergence issues and numerical stability, it is impractical to match the entire trajectory in one go; typically, a segment is sampled for matching at each iteration. However, previous TM-based methods overlook the potential interactions between matching different segments, particularly the presence of negative correlations. To study this problem, we conduct a quantitative analysis of the correlation between matching different segments and discover varying degrees of negative correlation depending on the image per class (IPC). Such negative correlation could lead to an increase in accumulated trajectory error and transform trajectory matching into a continual learning paradigm, potentially causing catastrophic forgetting. To tackle this issue, we propose a concurrent learning-based trajectory matching that simultaneously matches multiple segments. Extensive experiments demonstrate that our method consistently surpasses previous TM-based methods on CIFAR-10, CIFAR-100, Tiny ImageNet, and ImageNet-1K.

## 1 Introduction

The increasing scale of data has significantly enhanced the performance of neural networks (Brown et al., 2020; Kaplan et al., 2020; Hoffmann et al., 2022). However, it remains an unresolved question whether networks trained on much smaller datasets can achieve similar success. To address this question, Dataset Distillation (DD) (Wang et al., 2018) has emerged as a prominent research area due to its straightforward concept of distilling large datasets into smaller synthetic ones, while still maintaining comparable model performance. (Zhao et al., 2021; Cazenavette et al., 2022; Wang et al., 2022; Kim et al., 2022; Zhang et al., 2023). Among various data distillation methods (Zhao et al., 2021; Kim et al., 2022; Wang et al., 2022; Zhao & Bilen, 2023), Trajectory Matching (TM)-based methods (Cazenavette et al., 2022; Zhang et al., 2023; Guo et al., 2023) achieve excellent and even lossless results (Guo et al., 2023) by ensuring that the training trajectories on synthetic dataset closely match those of the full dataset. During the matching process, the complete training trajectory is divided into several segments for individual matching to ensure training stability and convergence (Cazenavette et al., 2022; Zhang et al., 2023; Guo et al., 2023).

However, this segmented matching scheme overlooks a critical issue: Matching different segments may be negatively correlated. This issue may bring an obstacle in the optimization because matching one segment can significantly increase the matching loss of other segments.

In this paper, we conduct an in-depth study on the correlation between different segments of trajectory matching. Specifically, we theoretically analyze how negative correlation affects the accumulated trajectory matching error (Du et al., 2023), and then we conduct a series of experiments to verify that negative correlations do exist. We monitor the matching loss of other epochs when one epoch is selected for matching. By calculating the Pearson correlation coefficient between the loss of the

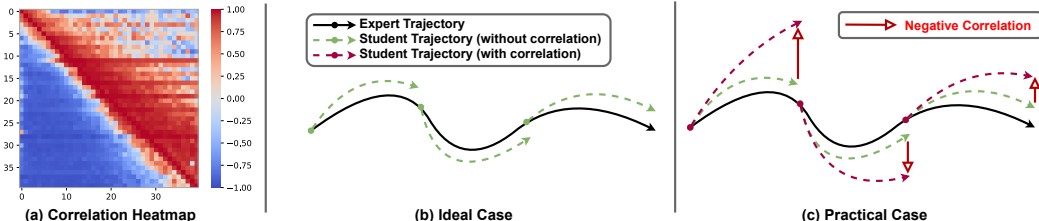

Figure 1: **(a)** Heatmap of the Pearson correlation coefficients (PCC): The element $(i, j)$ in the heatmap represents the PCC between the matching losses of epoch $j$ and the matching loss of epoch $i$, when epoch $i$ is the one being matched. It is evident that matching later epochs negatively correlates with earlier epochs, meaning that as the loss of later epochs decreases, the matching losses of earlier epochs increase. **(b)** In an ideal situation, the matching of each segment would not negatively affect the others. As long as each segment is accurately matched, the training trajectory on synthetic data can closely approximate the expert trajectory. **(c)** However, due to the negative correlation with other parts, matching other parts can cause it to deviate from the expert trajectory.

matched epoch and the losses of the unmatched epochs, we demonstrate the correlation between different segments. As shown in Figure 1 (a), the lower triangular portion of the heatmap matrix is predominantly negatively correlated, indicating that matching later parts of the trajectory significantly increases the loss of earlier parts. Moreover, we observe that the correlation between different segments also varies with the information capacity of the synthetic data, namely the Image per Class (IPC). When IPC is small, matching a late part exhibits a negative correlation with an early part, whereas at a large IPC, the negative correlation shifts to the upper triangular of the heatmap matrix. To understand this observation, we formalize the correlation and sampling strategy in the trajectory matching into a continual learning problem (Kirkpatrick et al., 2017; Chen & Liu, 2018; Kudithipudi et al., 2022), where matching different segments of the complete trajectory without strict correlation can lead to catastrophic forgetting (McClelland et al., 1995; McCloskey & Cohen, 1989). This makes the Existing TM-based methods unlikely to achieve a training trajectory on synthetic data that closely resembles the real expert trajectory.

To overcome this issue, we develop a Concurrent Training-based Trajectory Matching(ConTra) method. In the continual learning community, it is commonly believed that simultaneous multi-task learning (MTL) achieves optimal results when dealing with multiple negatively correlated tasks, representing an upper bound. Conversely, naive sequential learning (SL) is considered as a lower bound (Kirkpatrick et al., 2017; Shin et al., 2017; Schwarz et al., 2018). Therefore, instead of sampling a specific part from the complete trajectory to match each time as naive sequential learning (SL), we concurrently match those negatively correlated parts with multi-task learning (MTL). Furthermore, considering that different IPCs have varying information capacities, we employ a curriculum learning approach (Bengio et al., 2009a; Zhang et al., 2024) to generate the expert trajectory. Our experiments demonstrate that ConTra can consistently outperform other trajectory matching methods on CIFAR-10, CIFAR-100, Tiny ImageNet, and ImageNet-1K.

**Contributions.** **(1)** We theoretically analyze how the negative correlations affect the accumulated trajectory error and systematically quantify the correlation between matching different parts of a complete trajectory under various IPCs. **(2)** We explicitly highlight the inherent continual learning nature and the issue of catastrophic forgetting and based on this perspective, propose a new matching strategy—concurrent training—from the upper bound of continual learning, MTL.**(3)** We validate the effectiveness of our approach through extensive experiments.

## 2 RELATED WORK

(Wang et al., 2018) firstly formalized the concept of Dataset Distillation as a bi-level optimization problem, with the goal of distilling large-scale datasets into smaller synthetic ones while preserving comparable test performance. Dataset distillation can primarily be divided into following categories:

**Gradient matching.** Zhao et al. (2021) pioneered the gradient matching approach to Dataset Condensation (DC), which optimizes the synthetic data by minimizing difference between model gradients trained with a large training set and with the synthetic dataset. Kim et al. (2022) and Zhang et al. (2023) improved gradient matching by focusing on data regularity characteristics and model augmentation. MTT (Cazenavette et al., 2022) introduced a long-range matching strategy. FTD (Du et al., 2023) leveraged a flatter expert trajectory, and DATM (Guo et al., 2023) firstly achieved lossless condensation and conducted coarse-grained studies on matching early and late parts. PDD (Chen et al., 2023) generates several subsets to capture the entire training dynamics. However, all of the previous works used a segmented matching strategy and there is no detailed analysis of whether the matching of different segments is correlated.

**Distribution matching.** Another line of DD is feature or distribution matching, aiming at synthesizing data that can accurately approximate the distribution of the real training data(Wang et al., 2022; Zhao & Bilen, 2023; Zhao et al., 2023). They can only continually approximate lossless test accuracy and cannot achieve it with relatively small IPCs due to their spirit akin to coreset selection (Sener & Savarese, 2018; Welling, 2009)

**Kernel-based methods.** KIP (Nguyen et al., 2020), the first Kernel-based method, simplified dataset distillation into a single-level optimization problem through kernel ridge-regression with NTK (Lee et al., 2019). The computation cost of KIP scales quadratically in the number of pixels for convolutional kernels. Although subsequent studies (Zhou et al., 2022; Loo et al., 2023) have significantly reduced training costs, they still struggle to scale up to larger datasets and IPCs.

## 3 PRELIMINARIES

Let $\mathcal{T} = \{(x_i, y_i)\}_{i=1}^{|\mathcal{T}|}$ be a dataset with $|\mathcal{T}|$ samples, where $x_i \in \mathbb{R}^d$ and $y_i \in \mathcal{Y} = \{0, 1, \ldots, C-1\}$ are the input datapoint and its corresponding label, and $C$ is the number of classes. Dataset distillation aims to distill $\mathcal{T}$ into a much smaller synthetic dataset $\mathcal{S} = \{(s_i, y_i)\}_{i=1}^{|\mathcal{S}|}$, such that a model $f$ trained on the synthetic dataset $\mathcal{S}$ can achieve a comparable performance with a significant less training cost.

**Trajectory matching.** Trajectory matching (TM)-based methods achieve this goal by making the trajectories of models trained on synthetic dataset imitate the expert trajectories that are obtained on real dataset. Specifically, an expert trajectory $\tau^*$ is composed of a sequence of parameters that are partitioned into $T$ segments $\tau^* = \{\Theta_t^*\}_{t=0}^{T-1}$, and each segment $\Theta_t^* = (\theta_{t,0}^*, \theta_{t,1}^*, \ldots, \theta_{t,M}^*)$, where $M$ is a hyper-parameter that represents the length of segments. Several models are initialized and trained on the real dataset to get an expert trajectory set, $\{\tau^*\}$. In each iteration, a trajectory is sampled from $\{\tau^*\}$, and a segment of it, $\Theta_t^*$, is used for matching. During distillation, the start parameters of the student trajectory $\hat{\theta}_{t,0}$ are initialized with $\theta_{t,0}^*$ and then updated on the synthetic dataset for $N$ steps:

$$\hat{\theta}_{t,i+1} = \hat{\theta}_{t,i} - \alpha \nabla \ell \left( \mathcal{A}\left(b_{t,i}\right); \hat{\theta}_{t,i} \right), \text{ where } \hat{\theta}_{t,0} = \theta_{t,0}^*. \tag{1}$$

$\alpha$ is a learnable learning rate, $\mathcal{A}$ denotes differentiable augmentation function, and $b_{t+i}$ is the mini-batch sampled from $\mathcal{S}$. We aim for the student trajectory to closely approximate the actual trajectory after $N$ steps of updates. Formally, the matching loss is defined as follows:

$$\mathcal{L} = \frac{\left\| \hat{\theta}_{t,N} - \theta_{t,M}^* \right\|_2^2}{\left\| \theta_{t,0}^* - \theta_{t,M}^* \right\|_2^2}. \tag{2}$$

Subsequently, the synthetic dataset $\mathcal{S}$ is optimized by minimizing the matching loss of the segment, and this process of sampling a segment and then matching it is repeated multiple times to finally obtain a well-distilled dataset.

# 4 INCONSISTENT CORRELATIONS BETWEEN SEGMENTS MATCHING

Previous TM-based methods calculate the matching loss $\mathcal{L}$ by sampling a segment from the expert trajectory in each iteration. This paradigm assumes that if each segment of the trajectory is well-matched, the complete trajectory will also be matched accurately. However, this assumption is questionable. We find that if negative correlation exists between different segments, reducing the matching loss of a single segment may cause the complete trajectory to deviate from the real trajectory. In this section, we begin by demonstrating this issue from the perspective of accumulated trajectory error as introduced in (Du et al., 2023). We then empirically verify that negative correlations do exist prevalently in commonly used datasets.

## 4.1 THE IMPACT OF NEGATIVE CORRELATION ON ACCUMULATED TRAJECTORY ERROR

The ultimate goal of trajectory matching is to align complete trajectories trained on synthetic datasets with those from real datasets. To analyze the impact of negative correlation on this objective, we employ the accumulated matching error proposed in (Du et al., 2023) as a theoretical tool, which is used to measure the difference between in model parameters' weights obtained when training the model on the real training set versus the synthetic dataset during the **evaluation phase** (the synthetic dataset is already obtained by trajectory matching).

**Definition 1.** *Accumulated error. Let $\epsilon_t$ represent the accumulated trajectory error in the $t^{th}$ segment, which is defined as:*

$$\epsilon_t = \hat{\theta}_{t+1,0} - \theta^*_{t+1,0} = \hat{\theta}_{t,N} - \theta^*_{t,M}, \tag{3}$$

where $\hat{\theta}_{t,N}$ represents the final sets of parameters of the $t^{\text{th}}$ trajectory segment obtained on the synthetic dataset, which is also the initial parameters for the subsequent segment, i.e., $\hat{\theta}_{t,N} = \hat{\theta}_{t+1,0}$. Importantly, during evaluation, $\hat{\theta}_{t,0}$ is no longer initialized with $\theta^*_{t,0}$ and is continually updated by $\mathcal{S}$. Therefore, it is equal to the last set of weights in the previous segment, namely $\hat{\theta}_{t,0} = \hat{\theta}_{t-1,N}$.

The accumulated trajectory error of the last segment determines the final distance between the training trajectory on the synthetic dataset and the real trajectory. To analyze this more specifically, we introduce two additional error terms as followed:

**Definition 2.** *Initialization error. During training, the model for the $(t)^{th}$ segment is initialized with $\theta^*_{t,0}$, but in the evaluation phase, it is initialized with $\hat{\theta}_{t,0}$, which equals to $\theta^*_{t,0} + \epsilon_{t-1}$. This inconsistency incurs further discrepancies in the weights after subsequent gradient descent updates, namely the initialization error $\mathcal{I}$:*

$$\mathcal{I}_t = \mathcal{U}_{\mathcal{S}}(f_{\theta^*_{t,0}+\epsilon_{t-1}}, N) - \mathcal{U}_{\mathcal{S}}(f_{\theta^*_{t,0}}, N), \tag{4}$$

where $\mathcal{U}_{\mathcal{S}}(f_\theta, N)$ denotes the updates of model $f$ after $N$ steps gradient decent on the synthetic dataset $\mathcal{S}$, starting with parameter $\theta$.

**Definition 3.** *Matching error represents the distance between the endpoint of the sampled segment that we try to minimize during optimizing the synthetic dataset in distillation step The matching error of the $(t)^{th}$ is defined as followed:*

$$\delta_t = (\mathcal{U}_{\mathcal{S}}(f_{\theta^*_{t,0}}, N) - \mathcal{U}_{\mathcal{T}}(f_{\theta^*_{t,0}}, M)) \tag{5}$$

Then we have:

**Theorem 1.** *Assuming there are $T$ segments in total, the accumulated error of the last segment is the sum of the matching errors and the initialization errors from all preceding segments:*

$$\epsilon_{T-1} = \sum_{i=1}^{T-1} \mathcal{I}_i + \sum_{i=0}^{T-1} \delta_i, \text{ where } \delta_0 = \epsilon_0. \tag{6}$$

The proof of Theorem. 1 is provided in Appendix A.1. Previous TM-based methods sample only one segment to minimize the matching loss as described in Equation 2, essentially involving the random selection of a $\delta_i$ to minimize. However, when the minimization of the matching error for

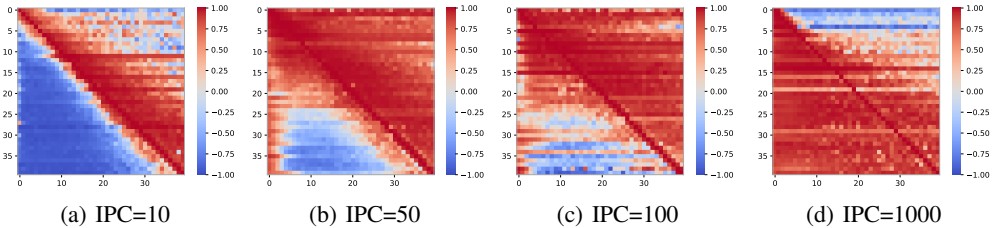

(a) IPC=10            (b) IPC=50            (c) IPC=100            (d) IPC=1000

Figure 2: Heatmap of the Pearson correlation coefficients (PCC) on CIFAR-10: The element $(i, j)$ in the heatmap represents the PCC between the matching losses of epoch $j$ and the matching loss of epoch $i$, when epoch $i$ is the one being matched.

different segments is negatively correlated, reducing the $\delta_i$ of one segment may lead to an increase in the matching loss of other segments $\sum_{j \neq i} \delta_j$. Consequently, rather than decrease, $\epsilon_{T-1}$ might actually increase. This phenomenon makes it challenging for the trajectory trained from synthetic data to closely approximate the exact trajectory, irrespective of the addition of more segments.

## 4.2 Matching Different Segments Exhibits Negative Correlation (Empirically)

Based on the analysis presented in Section 4.1, we clarify the impact of negative correlation on trajectory matching. In this subsection, we conduct experiments to verify the prevalence of negative correlation when matching different segments. We leverage the Pearson Correlation Coefficient (PCC) for a quantitative analysis of the correlation. For simplicity, our experiments are conducted on CIFAR-10 with a complete training trajectory comprises 40 epochs where each representing a segment with multiple checkpoints. We first establish a complete $\tau_o$ as the reference trajectory, which will not participate in the distillation process. For each iteration, we sample a trajectory and match a fixed part of it—specifically, 1 of the 40 epochs. Subsequently, we monitor the changes in matching losses (Eq. 2) for the matched epoch and the remaining 39 epochs on the trajectory $\tau_o$. Specifically, when matching the $i^{\text{th}}$ epoch, the PCC between the matching loss of the $i^{\text{th}}$ epoch and the $j^{\text{th}}$ epoch is defined as:

$$r_{ij} = \frac{\sum_{z=1}^{Z} \left( \mathcal{L}_{i,z} - \bar{\mathcal{L}}_i \right) \left( \mathcal{L}_{j,z} - \bar{\mathcal{L}}_j \right)}{\sqrt{\sum_{z=1}^{Z} \left( \mathcal{L}_{i,z} - \bar{\mathcal{L}}_i \right)^2} \sqrt{\sum_{z=1}^{Z} \left( \mathcal{L}_{j,z} - \bar{\mathcal{L}}_j \right)^2}}, \tag{7}$$

where $Z$ denotes the total number of distillation iterations, $\mathcal{L}_{i,z}$ is the matching loss of the $i^{\text{th}}$ segment (epoch) of $\tau_o$ during the $z^{\text{th}}$ iteration. The PCC is positive if $\mathcal{L}_i$ and $\mathcal{L}_j$ trends both decrease or increase simultaneously. Conversely, the PCC is negative when the trends of $\mathcal{L}_i$ and $\mathcal{L}_j$ diverge.

**Negative correlation exists prevalently.** Figure 2 shows the heatmaps of PCCs under different IPCs. When the IPC is relatively small, matching later parts exhibits a strong negative correlation with earlier parts, with negative correlations concentrated in the lower triangular area of the heatmap. This pattern highlights a significant issue of matching later segments while earlier segments are forgotten. In practical implementation of previous work, when the IPC is 10, it is common that segments are only sampled from the first 20 epochs for matching (Cazenavette et al., 2022; Cui et al., 2023; Guo et al., 2023). Therefore, sampling later segments clearly leads to a deviation of the previously well-matched early part from the real trajectory.

As the IPC increases, the negatively correlated parts gradually shift from the lower triangular area to the upper triangular area. When IPC reaches 1000, matching the early part causes an increase in the matching loss of the well-matched late part. Experiments in (Guo et al., 2023) demonstrate that at an IPC of 1000, matching only the late part yields satisfactory outcomes. From a correlation perspective, this is because, at this IPC level, matching the late part is positively correlated with matching the entire trajectory.

**Roles of Training dynamics in the correlation variation.** Although neural networks can nearly memorize the entire training set (Zhang et al., 2021), the fitting of samples is a dynamic process. In the early epochs of training, those easy patterns (Carlini et al., 2022) dominate the matching

gradient (Arpit et al., 2017). That is to say, conducting gradient matching in the early epochs causes the synthetic data to primarily fit the easy patterns, such as lines and curves. In the late stages of model training, the situation becomes more complex. *In particular, the training process does not exclusively focus on fitting hard patterns in later stages*; rather, it dynamically adjusts to also refit simpler ones as needed (Arpit et al., 2017; Katharopoulos & Fleuret, 2018). This dynamic process is controlled by both easy and hard patterns, and it thus contains the information of the entire dataset.

Regarding the variation in correlation with changes in IPC, we speculate that with a small IPC, the synthetic dataset's limited capacity suffices only to fit simple patterns. Easy patterns learned through matching early segments are likely forgotten when matching later segments, so matching late segments are negatively correlated with early ones. In contrast, a high IPC enables the synthetic dataset to simultaneously fit both simple and complex patterns through complex training dynamics, facilitating lossless distillation as reported in (Guo et al., 2023). With this increased IPC, matching early epochs may result in the loss of information regarding complex patterns learned in later segments, thereby exhibiting a negative correlation with these later epochs.

## 5 CONCURRENT TRAINING-BASED TRAJECTORY MATCHING

**Trajectory matching as a continual learning problem.** As discussed in Section 4, it is evident that various correlations exist between different segments' matching, resembling the scenario in continual learning where different tasks exhibit high diversity. In continual learning, when tasks are either uncorrelated or negatively correlated, sequential learning—where tasks are optimized one by one—often leads to the phenomenon of catastrophic forgetting. This phenomenon occurs when adaptation to a new task significantly diminishes the ability to perform previous tasks(Wang et al., 2023). This parallel is precisely what we have observed in trajectory matching, where we regard the matching of different segments as separate tasks. According to Eq. 6, our objective is to minimize cumulative matching errors across different segments, representing the aggregated performance across all tasks. Previous strategies failed to consider the potential for catastrophic forgetting by employing a naive sequential learning (SL), which minimizes the performance of each task, i.e., $\delta$, sequentially. However, SL is considered the least effective learning paradigm, thereby serving as a lower bound in continual learning (Kirkpatrick et al., 2017; Shin et al., 2017; Schwarz et al., 2018).

**Concurrent training.** For multiple negatively correlated tasks, compared to naive SL, a simple yet effective method to significantly enhance the aggregated performance of these tasks is to learn them simultaneously. This approach, known as multi-task learning (MTL) or concurrent training, is considered the upper bound in continual learning (Kirkpatrick et al., 2017; Shin et al., 2017; Schwarz et al., 2018).

Specifically, suppose a complete expert trajectory $\tau^*$ comprises $T$ segments. Previous methods typically set an upper bound $T^+$ and a lower bound $T^-$, and only one segment within this range $\{\Theta^*_{T^-}, \cdots, \Theta^*_{T^+}\}$ is sampled to match (Guo et al., 2023; Cazenavette et al., 2022; Du et al., 2023). Compared to them, in addition to sampling, we match multiple segments within this range simultaneously, and the objective is defined as:

$$\mathcal{L} = \beta \frac{\left\| \hat{\theta}_{t,N} - \theta^*_{t,M} \right\|_2^2}{\left\| \theta^*_{t,0} - \theta^*_{t,M} \right\|_2^2} + (1-\beta) \sum_{i=0}^{K-1} \frac{1}{K} \frac{\left\| \hat{\theta}_{T^-+iR,N} - \theta^*_{T^-+iR,M} \right\|_2^2}{\left\| \theta^*_{T^-+iR,0} - \theta^*_{T^-+iR,M} \right\|_2^2} \tag{8}$$

where $\beta$ is the coefficient to balance the sampling and concurrent training, $K$ represents the number of tasks, which corresponds to the number of segments matched simultaneously, and $R$ is the distance between each segment that are simultaneously matched. $\hat{\theta}_{T^-+iR,N}$ is obtained by $N$ steps optimization on $\hat{\theta}_{T^-+iR,0}$ on the synthetic dataset, where $\hat{\theta}_{T^-+iR,0}$ is the starting parameters of the segment $i$ and $\hat{\theta}_{T^-+iR,0} = \theta^*_{T^-+iR,0}$. There are two notable points here. First, the segments are not necessarily consecutive, namely $R$ could be larger than the length of a segment. Figure 2 suggests that matching one segment is often positively correlated with matching adjacent segments. Thus, as long as the gaps between non-consecutive segments are not too large, their matching loss will also decrease in tandem with the decrease in matching loss of adjacent segments. Second, $T^- + (K-1)R$ is close to $T^+$, ensuring that the entire trajectory within the range are matched.

Table 1: Comparing with previous dataset distillation methods on CIFAR-10, CIFAR-100, and Tiny ImageNet. Both distillation and evaluation use ConvNETs, with the best results highlighted in bold.
[1] For FTD, we followed the settings from (Guo et al., 2023), removing the exponential moving average.
[2] PDD (Chen et al., 2023) is a plug-in module which can be combined with any TM-base methods; Here is the experimental results of PDD+MTT.
[3] Previous TM-based methods worse than random initialization in higher IPC are indicated by ↘.

| Dataset | CIFAR-10 | | | | | CIFAR-100 | | | | Tiny ImageNet | | |
|---|---|---|---|---|---|---|---|---|---|---|---|---|
| IPC | 1 | 10 | 50 | 500 | 1000 | 1 | 10 | 50 | 100 | 1 | 10 | 50 |
| Ratio | 0.02 | 0.2 | 1 | 10 | 20 | 0.2 | 2 | 10 | 20 | 0.2 | 2 | 10 |
| Random | 14.4±2.0 | 26.0±1.2 | 43.4±1.0 | 73.2±0.3 | 78.4±0.2 | 4.2±0.3 | 14.6±0.5 | 30.0±0.4 | 42.8±0.3 | 1.4±0.1 | 5.0±0.2 | 15.0±0.4 |
| DC | 28.3±0.5 | 44.9±0.5 | 53.9±0.5 | 72.1±0.4 | 76.6±0.3 | 12.8±0.3 | 25.2±0.3 | - | - | - | - | - |
| DM | 26.0±0.8 | 48.9±0.6 | 63.0±0.4 | 75.1±0.3 | 78.8±0.1 | 11.4±0.3 | 29.7±0.3 | 43.6±0.4 | - | 3.9±0.2 | 12.9±0.4 | 24.1±0.3 |
| DSA | 28.8±0.7 | 52.1±0.5 | 60.6±0.5 | 73.6±0.3 | 78.7±0.3 | 13.9±0.3 | 32.3±0.3 | 42.8±0.4 | - | - | - | - |
| CAFE | 30.3±1.1 | 46.3±0.6 | 55.5±0.6 | - | - | 12.9±0.3 | 27.8±0.3 | 37.9±0.3 | - | - | - | - |
| KIP | 49.9±0.2 | 62.7±0.3 | 68.6±0.2 | | | 15.7±0.2 | 28.3±0.1 | - | - | - | - | - |
| MTT[3] | 46.2±0.8 | 65.4±0.7 | 71.6±0.2 | ↘ | ↘ | 24.3±0.3 | 39.7±0.4 | 47.7±0.2 | 49.2±0.4 | 8.8±0.1 | 23.2±0.2 | 28.0±0.3 |
| FTD[1,3] | 46.0±0.4 | 65.1±0.4 | 73.2±0.2 | ↘ | ↘ | 24.4±0.4 | 42.5±0.2 | 48.5±0.3 | 49.7±0.4 | 10.5±0.2 | 23.4±0.3 | 28.2±0.4 |
| TESLA[3] | 48.5±0.8 | 66.4±0.8 | 72.6±0.7 | ↘ | ↘ | 24.8±0.4 | 41.7±0.3 | 47.9±0.3 | 49.2±0.4 | - | - | - |
| PDD[2] | - | 66.9±0.4 | 74.2±0.5 | - | - | - | 43.1±0.7 | 52.0±0.5 | - | - | 27.3±0.5 | 29.2±0.6 |
| DATM | 46.9±0.5 | 66.8±0.2 | 76.1±0.3 | 83.5±0.2 | 85.5±0.4 | 27.9±0.2 | 47.2±0.4 | 55.0±0.2 | 57.5±0.2 | 17.1±0.3 | 31.1±0.3 | 39.7±0.3 |
| ConTra | **50.0±0.6** | **68.3±0.4** | **76.9±0.4** | **84.0±0.1** | **86.1±0.2** | **28.5±0.3** | **48.9±0.2** | **55.5±0.2** | **58.0±0.1** | **17.7±0.2** | **32.9±0.4** | **40.2 ±0.2** |
| Full Data | 84.8±0.1 | | | | | 56.2±0.3 | | | | 37.6±0.4 | | |

In each iteration, we choose $K$ segments and 1 randomly sampled segment from an expert trajectory to match, and then optimize the synthetic dataset by performing back-propagation with respect to the matching loss Eq. 8. The whole algorithm is provided in Appendix C.

During our experiments, we also tried some techniques used in continual learning, such as Synaptic Intelligence (SI) (Zenke et al., 2017) and Elastic Weight Consolidation (EWC) (Kirkpatrick et al., 2017). They do bring some improvements, but none are as simple and effective as directly conducting concurrent training for multiple tasks.

**Information capacity.**    According to the analysis in Section 4.1, information capacity is a crucial factor that influences the correlation between matched segments, especially when the capacity is extremely limited, such as IPC = 1 or 10. Therefore, we should prioritize learning as many easy patterns as possible. Therefore, we leverage a curriculum learning approach (Bengio et al., 2009b; Zhang et al., 2024) to generate the expert trajectories, ensuring that the early part of the trajectory primarily fits samples that can be easily classified. We only use curriculum learning with very limited capacity, such as when the IPC is 1 and 10. For the details of this trick, please refer to Appendix B.

## 6 EXPERIMENTS

### 6.1 SETUP

**Datasets and models.**    Following recent work (Guo et al., 2023; Liu et al., 2023), we conduct experiments on several popular datasets, including CIFAR-10, CIFAR-100 (Krizhevsky et al., 2009), and Tiny-imagenet (Le & Yang, 2015). Following previous works (Zhao et al., 2021; Cazenavette et al., 2022; Guo et al., 2023), unless specified otherwise, both distillation and evaluation utilize a 3-layer convolutional network, while Tiny-imagenet employs a 4-layer configuration. We adopt the differentiable augmentation widely used in previous work (Cazenavette et al., 2022; Guo et al., 2023; Du et al., 2023; Cui et al., 2023). We also use the soft label and initialization with correct samples introduced in (Guo et al., 2023). We provide more details in Appendix D.

**Baselines.**    To verify the efficacy of our method, we compare it with some popular baselines and State-of-The-Art methods, including DC (Zhao et al., 2021), DM (Zhao & Bilen, 2023), DSA (Zhao & Bilen, 2021), CAFE (Wang et al., 2022), KIP (Nguyen et al., 2020), MTT (Cazenavette et al., 2022), FTD (Du et al., 2023), TESLA (Cui et al., 2023), PDD (Chen et al., 2023), and DATM (Guo et al., 2023). Kenel-based methods (Nguyen et al., 2020; Zhou et al., 2022; Loo et al., 2023) use a ConvNet of much larger width (1024, other methods are 128), so we only choose KIP as the baseline. Top-1 accuracy is the main metric to evaluate the distillation's performance.

## 6.2 Comparison with State-of-The-Art Methods

Table 1 presents the mean and standard deviation of 5 runs for various dataset distillation methods on CIFAR-10, CIFAR-100, and Tiny ImageNet. We observe that ConTra consistently outperforms the baselines across different IPCs, especially when the information capacity is limited, i.e., when the IPC is small. Specifically, ConTra surpasses DATM by margins of 3.1%/1.5% on CIFAR-10 with IPC 1/10. In such cases, the synthetic dataset is insufficient to capture the complex training dynamics, leading to strong negative correlations between matching different segments. Matching later segments can increase the matching loss of previously matched earlier segments.

Table 2: Cross-architecture generalization: Test performance of other representative models trained on the synthetic dataset distilled through ConvNet. We highlight the best performance in bold.

| IPC | Method | ResNet18 | AlexNet | VGG11 |
|-----|--------|----------|---------|-------|
| 10 | MTT | 45.7±0.8 | 34.0±1.9 | 50.2±0.5 |
| | PDD | 43.5±0.6 | 18.3±1.5 | 44.0±0.6 |
| | DATM | 47.7±0.4 | 38.8±0.8 | 46.1±0.6 |
| | ConTra | **52.9±0.5** | **42.4±1.3** | **50.6±0.3** |
| 50 | MTT | 62.9±0.3 | 51.1±1.2 | 57.5±0.8 |
| | PDD | 60.5±0.5 | 16.3±2.2 | 48.2±0.5 |
| | DATM | 65.9±0.8 | 53.4±1.6 | 60.1±0.4 |
| | ConTra | **66.2±0.3** | **56.0±1.5** | **62.5±0.4** |

Concurrent training effectively alleviates this issue,and meanwhile, the curriculum training enables the synthetic dataset to focus on simple patterns and samples that are easy to fit. Another notable point is that ConTra achieve lossless condensation with a 20% ratio on CIFAR-10 and CIFAR-100, and a 10% ratio on Tiny ImageNet.

## 6.3 Cross-Architecture Generalization

The process of dataset distillation is conducted on a specific model. Therefore, it is crucial to verify whether the synthetic dataset distilled through a single model can be applied effectively to other models. Table 2 shows the test accuracy of other models trained on the synthetic dataset distilled by ConvNet on CIFAR-10. We can observe that whether the IPC is 10 or 50, compared to other TM-based methods, ConTra achieves the best performance across several popular models.

## 6.4 Ablation Study

**Concurrent training: a plug-in module.** Concurrent training, the core component of our method, is an plug-in module that can be integrated with any trajectory matching method. By simply replacing the sampling loss in Eq. 2 with the loss Eq. 8, previous TM-based methods can be adapted to operate in a concurrent training mode. To validate the efficacy of concurrent training, we incorporate it with MTT and DATM which are the vanilla TM method ans SOTA respectively, and the results are presented in Table 3.

Table 3: The performance of two representative TM-based methods MTT and DATM combined with concurrent training.

| Dataset | CIFAR-10 | | CIFAR-100 | |
|---------|----------|------|-----------|------|
| IPC | 1 | 10 | 1 | 10 |
| MTT | 46.2 | 65.4 | 24.3 | 39.7 |
| MTT+CT | **48.1** | **67.1** | **26.0** | **43.3** |
| DATM | 46.9 | 66.8 | 27.9 | 47.2 |
| DATM+CT | **48.5** | **67.9** | **28.2** | **48.6** |

We can see concurrent training significantly enhances both MTT and DATM. Specifically, MTT improves by 1.1% to 3.6%, while DATM increases by 0.3% to 1.6%. This demonstrates that concurrent training can serve as a versatile module, capable of combining with other trajectory matching methods. Additionally, the improvements are more pronounced when the IPC is smaller, and with more substantial gains on CIFAR-10 compared to CIFAR-100. This further confirms that when the information capacity is lower, the negative correlation between matching different segments is more significant, making multi-task training more effective.

**How does concurrent training affect correlation?** To verify that ConTra indeed alleviates the negative correlation problem, we conduct the experiments described in Section 4.2, displaying heatmaps of the Pearson correlations coefficients between matching different segments. Notably that

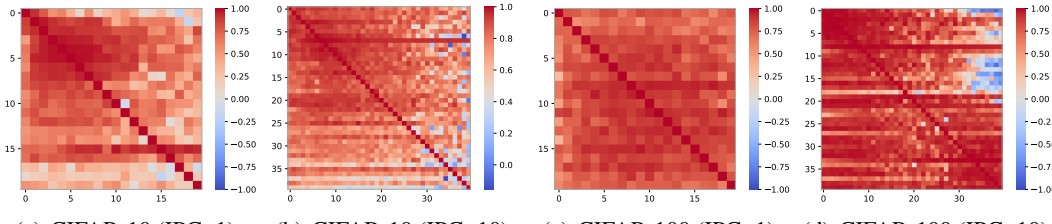

(a) CIFAR-10 (IPC=1)    (b) CIFAR-10 (IPC=10)    (c) CIFAR-100 (IPC=1)    (d) CIFAR-100 (IPC=10)

Figure 3: Heatmap of the Pearson correlation coefficients on CIFAR-10 and CIFAR-100, with IPC=1 and IPC=10.

in Section 4.2, we set up the experiments to match only a specific segment, whereas for ConTra, we applied the same setup to the sampling, with concurrent training still matching multiple segments within the range. We select IPC=1 and IPC=10, where negative correlation is most severe for demonstration. For IPC=1, we only used the first 20 epochs due to the extremely low information capacity; no trajectory matching method utilize the trajectories beyond 20 epochs. The results are shown in Figure 3, where it can be observed that ConTra's concurrent training strategy exhibits positive correlations across nearly all segments. The results on CIFAR-100 exhibits stronger positive correlations compared to CIFAR-10 because, at the same IPC, the size of the synthetic dataset for CIFAR-100 is ten times that of CIFAR-10, providing a larger information capacity.

**Number of tasks.** The number of tasks $K$ represents the number of segments matched simultaneously in Eq. 8. These segments should be evenly distributed between the lower bound $T^-$ and the upper bound $T^+$, so the range $R$ is set to $\lfloor (T^+ - T^+)/K \rfloor$. To explore the impact of $K$, we conduct experiments on CIFAR-10 and set the number of tasks from 2 to 6. The performance of our method and vanilla MTT with concurrent training (MTT+CT) is shown in Figure 4 (left). As $K$ rises, the concurrent training brings non-trivial improvement on MTT and our method. We notice that the improvements brought by increasing $K$ gradually slow down as $K$ continues to grow. We speculate that this is because, as $K$ increases, the distance $R$ between segments decreases, and Figure 2 indicates that closer segments exhibit more positive correlation. When $K$ is sufficiently large, every part of the full trajectory can find a matching segment that is positively correlated with it, making further increase $K$ less effective.

**Curriculum learning.** Curriculum learning is not a primary contribution of this work. We only use it as a trick with very low IPC, such as when the IPC is 1 and 10 in CIFAR10. We provide the ablation study in Table 4 left, showing that focusing on easy patterns can bring some performance improvement when the information capacity is extremely low.

Table 4: **Left**: The performance of ConTra with and without curriculum learning. **Right**: We present the number of iterations required to converge (approximately) and the training time for various values of $K$ (number of tasks) on the NVIDIA H800 GPU (IPC=10), measured in hours per 1000 iterations.

| Dataset | CIFAR-10 | | CIFAR-100 | |
|---|---|---|---|---|
| IPC | 10 | 100 | 10 | 100 |
| ConTra | **50.0** | **68.3** | **28.5** | **48.9** |
| ConTra w/o CL | 48.3 | 67.6 | 28.1 | 48.6 |

| Method | DATM | $K$=2 | $K$=3 | $K$=4 | $K$=5 |
|---|---|---|---|---|---|
| **CIFAR-10** | 0.32 | 0.57 | 0.87 | 1.10 | 1.31 |
| **CIFAR-100** | 1.64 | 3.38 | 5.15 | 6.91 | 8.36 |
| **# of iters** | 4000 | 3100 | 2500 | 2000 | 1500 |

**Balance coefficient.** In Figure 4 (right), we investigate the impact of the balance coefficient, $\beta$, on the performance of ConTra. $\beta$ quantifies the reliance of ConTra on the sampling segment when computing the matching loss. To achieve optimal results, $\beta$ should not be too large, as a larger value of $\beta$ makes the approach more akin to traditional sampling-based trajectory matching methods.

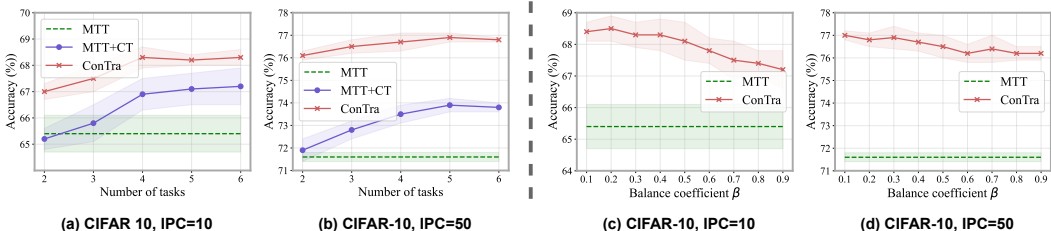

Figure 4: **Left**:Mean test accuracy and standard deviation of 5 runs on CIFAR-10 after training on the distilled dataset with different number of tasks in concurrent training. **Right:** :Mean test accuracy and standard deviation of 5 runs on CIFAR-10 after training on the distilled dataset with different balance coefficient $\beta$ in Eq. 8.

### 6.5 COST ANALYSIS

We report the training time for various values of $K$ on the NVIDIA H800 GPU (IPC=10) in Table 4 right, measured in hours per 1000 iterations. The time cost is approximately proportional to the value of $K$ (number of tasks), but we find that larger $K$ values lead to faster convergence. For example, on CIFAR-10, DATM converges at 4000 iterations, whereas ConTra with $K = 5$ requires only about 1500 iterations. ConTra does not incur additional GPU memory costs, as we can compute the gradient of different tasks and backpropagate them, separately. Despite ConTra is slower, this does not affect our core contribution: identifying the negative correlation in trajectory matching when matching different segments. Concurrent Training, as a straightforward solution, offers significant improvements.

### 6.6 ADDITIONAL EXPERIMENTS

**Stability and visualizations.** Another disadvantage of negative correlation is that it can cause training to be highly unstable and convergence to be poor. We demonstrate the superior stability of our method in Appendix E.1. In Appendix E.5, we provide visualizations of parts of the synthetic dataset.

**Scalability** We can scale up ConTra to ImageNet-1K using TESLA (Cui et al., 2023). TESLA is a plug-in trick that can compute the unrolled gradient in trajectory matching with constant memory complexity. The result is provided in Appendix E.2, which demonstrate that concurrent learning can also yield improvements on large-scale datasets.

**Generalization across other architectures.** To the best of our knowledge, cross-architecture generalization from CNNs to Transformer-based models remains an unexplored problem. We study the generalization VIT in Appendix E.4, and we find that trajectory matching is structurally bound; therefore, synthetic datasets distilled from CNNs struggle to achieve good generalization performance on ViTs.

**Downstream task.** We also perform experiments on neural architecture search, detailed in Appendix E.3. We implement NAS on CIFAR10 with the search space of 720 ConvNets varying in network depth, width, activation, normalization, and pooling. The result demonstrates that the synthetic datasets distilled by ConTra can perform well in downstream task.

## 7 CONCLUSION

In this work, we systematically study the interactions between matching different segments in trajectory matching. We further analyze the potential effect of the negative correlation from the perspectives of accumulated trajectory error and catastrophic forgetting and argue that such correlation cannot be ignored. Based on these analyses, we propose a simple yet effective method, ConTra, and validate its effectiveness through extensive experiments.

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

# A  PROOF

## A.1  PROOF OF THEOREM 1.

Firstly, we consider the accumulated error of $(t+1)^{th}$ segment $\epsilon_{t+1}$:

$$
\begin{aligned}
\epsilon_{t+1} =& \hat{\theta}_{t+2,0} - \theta^*_{t+2,0} = \hat{\theta}_{t+1,N} - \theta^*_{t+1,M} \\
=& (\hat{\theta}_{t+1,0} + \mathcal{U}_\mathcal{S}(f_{\theta^*_{t+1,0}+\epsilon_t}, N)) - (\theta^*_{t+1,0} + \mathcal{U}_\mathcal{T}(f_{\theta^*_{t+1,0}}, M)) \\
=& \epsilon_t + (\mathcal{U}_\mathcal{S}(f_{\theta^*_{t+1,0}+\epsilon_t}, N) - \mathcal{U}_\mathcal{S}(f_{\theta^*_{t+1,0}}, N)) + (\mathcal{U}_\mathcal{S}(f_{\theta^*_{t+1,0}}, N) - \mathcal{U}_\mathcal{T}(f_{\theta^*_{t+1,0}}, M)),
\end{aligned}
\tag{9}
$$

According to Definition. 1 and Definition. 2, $\mathcal{I}_{t+1} = \mathcal{U}_\mathcal{S}(f_{\theta^*_{t+1,0}+\epsilon_t}, N) - \mathcal{U}_\mathcal{S}(f_{\theta^*_{t+1,0}}, N))$, and $\delta_{t+1} = (\mathcal{U}_\mathcal{S}(f_{\theta^*_{t+1,0}}, N) - \mathcal{U}_\mathcal{T}(f_{\theta^*_{t+1,0}}, M))$. Then we have:

$$
\epsilon_{t+1} = \epsilon_t + \mathcal{I}_{t+1} + \delta_{t+1}.
\tag{10}
$$

$\delta_{t+1}$ is the *matching error* of segment $t+1$ that we try to minimize during optimizing the synthetic dataset in distillation step. Assuming there are $T$ segments in total, the final accumulated trajectory error, $\epsilon_{T-1}$, follows recursively that:

$$
\epsilon_{T-1} = \sum_{i=1}^{T-1} \mathcal{I}_i + \sum_{i=0}^{T-1} \delta_i, \text{ where } \delta_0 = \epsilon_0,
\tag{11}
$$

where $\mathcal{I}_0 = 0$ and $\delta_0 = \epsilon_0$ because there is no accumulated error before the first segment.

# B  CURRICULUM LEARNING

Zhang et al. (2024) tries to incorporate curriculum learning into the generation of expert trajectories in graph condensation task. Inspired by them, we prepare curriculum-based trajectory expert trajectory on image datasets. The core idea of curriculum learning is to arrange samples from simple to complex, allowing the model to mimic the human learning process by starting with simple samples and gradually progressing to more complex ones (Bengio et al., 2009b; Krueger & Dayan, 2009).

In TM-based distillation methods, the size of the IPC indicates the information capacity of the synthetic dataset. Therefore, when the IPC is small, it is crucial to focus more on the simple samples that constitute the majority of the real dataset. We define the learning difficulty of samples based on the order in which they are correctly classified during the model's training process on the real dataset. Samples that are classified correctly earlier are considered easy samples, while those classified correctly later are deemed more complex. After assigning sample difficulty, we sort the entire training set according to sample difficulty. Initially, the training set includes only simple samples; as training progresses, complex samples are incrementally introduced using a linear function. To manage this progression, we use a pacing function $h(e)$ that maps each training epoch $e$ to the proportion of samples selected from the ordered training set. The pacing function $h(e)$ is defined as follows:

$$
h(t) = min(1, \lambda + (1 - \lambda)\frac{e}{\gamma},
\tag{12}
$$

where $\lambda$ is the initial proportion of the training set, and $\gamma$ is the threshold of epoch when the full dataset is used. The expert trajectory obtained in this way ensures that early epochs mainly contains easy patterns. We only use this trick for low IPC experiments, as we find it doesn't work for IPC larger than 10.

# C  ALGORITHM

The algorithm of concurrent training is shown in Algorithm 1. In line 1-2, we initialize the synthetic dataset $\mathcal{S}$ from the real dataset $\mathcal{T}$. Line 3 to 20 are the distillation loop. In each iteration, we randomly sample an expert trajectory from $\{\tau^*\}$ (line 4) and sample one segment from it (line 5). Meanwhile, we choose $K$ segments with a distance $R$ between each from the expert trajectory (line 7). Then we

initialized a student networks for each segment (line 6 and 8 to 10). From line 11 to 17, we update the student networks on the synthetic dataset to get their parameters after $N$ steps. Finally, we compute the matching loss using Eq. 8 (line 18) and update the synthetic dataset and the learning rate of student networks by backpropagation (line 19).

---

**Algorithm 1:** Concurrent Training-based Trajectory Matching

---

**Input:** $\{\tau^*\}$: set of expert parameter trajectories obtained on $\mathcal{T}$.
**Input:** $M$: # length of each segment in the the expert trajectory.
**Input:** $N$: # update steps of student network per distillation iteration.
**Input:** $R$: distance between each segment.
**Input:** $\beta$: coefficient to balance the sampling and concurrent training.
**Input:** $T^- < T^+$: the lower and upper bound of the expert trajectory that used to match.
**Output:** The distilled dataset $\mathcal{S}$

1 Initialize distilled dataset $\mathcal{S} \sim \mathcal{T}$;
2 Initialize the learning rate $\alpha$ for training model on $\mathcal{S}$;
3 **for** *iter=1, ..., Iteration$_{max}$* **do**
4      Sample an expert trajectory $\tau^* \sim \{\tau^*\}$ with $\tau^* = \{\Theta_t^*\}_{t=0}^{T-1}$;
5      Sample a start point between $T^-$ and $T^+$;
6      Initialize a student network with expert params $\hat{\theta}_{t,0} := \theta_{t,0}^*$;
7      Choose $K$ segments within $T^-$ and $T^+$ with the distance $R$ between each of them;
8      **for** *i=0, ..., K − 1* **do**
9          Initialize a student network with expert params $\hat{\theta}_{T^-+iR,0} := \theta_{T^-+iR,0}^*$;
10      **end**
11      **for** *n=0, ..., N − 1* **do**
12          $b_{t,n} \sim \mathcal{S}$            ▷ Sample a mini-batch from distilled dataset;
13          $\hat{\theta}_{t,n+1} = \hat{\theta}_{t,n} - \alpha \nabla \ell \left( \mathcal{A}(b_{t,n}); \hat{\theta}_{t,n} \right)$       ▷ Update the model on $\mathcal{S}$;
14          **for** *i=0, ..., K − 1* **do**
15              $\hat{\theta}_{T^-+iR,n+1} = \hat{\theta}_{T^-+iR,n} - \alpha \nabla \ell \left( \mathcal{A}(b_{t,n}); \hat{\theta}_{T^-+iR,n} \right)$;
16          **end**
17      **end**
18      Compute the loss $\mathcal{L}$ using Eq. 8;
19      Update $\mathcal{S}$ and $\alpha$ with respect to $\mathcal{L}$;
20 **end**
21 return the distilled syntactic dataset $\mathcal{S}$;

---

# D    MORE DETAILS OF EXPERIMENTS

**Distillation settings.** Consistent with previous work (Cazenavette et al., 2022; Guo et al., 2023), we conduct 10000 iterations of distillation to ensure adequate convergence employ ZCA whitening as in all experiments as default (Nguyen et al., 2020; 2021).

**Evaluation settings.** Following previous methods (Cazenavette et al., 2022; Guo et al., 2023), we train a randomly initialized neural network on the synthetic dataset and then assess its performance on the validation set of the true dataset using the top-1 accuracy metric. All reported results represent the mean and standard deviation from 5 repeated runs. For performance of baseline in Table 1, we use results reported in their respective literature to ensure a fair comparison as done in previous work (Guo et al., 2023; Chen et al., 2023).

**Architecture.** We use the same network architecture as previous work (Cazenavette et al., 2022), a 3-layer ConvNet for CIFAR-10 and a 4-layer Convnet for Tiny ImageNet. Each layer of ConNet comprises a 128-kernel convolutional layer, an instance normalization layer (Ulyanov et al., 2016), a ReLU activation function, and an average pooling layer. Except for the cross-architecture

generalization experiments, the same network architecture is used for both distillation and evaluation in all other experiments.

**Computational resources.**   We conduct our experiments using 1-4 NVIDIA H800 GPUs. The number of GPUs utilized depends on the size of the dataset and the IPC. If computational resources are limited, employing techniques from TESLA (Cui et al., 2023) to reduce the storage of computational graphs can enable all experiments to be conducted on a single 80GB GPU.

**Hyper-parameters.**   We provide the hyper-parameters of our method in Table 5, where $R$ is the distance between the start point of each task and $K$ is the number of tasks. Notably, the segments are not necessarily consecutive, namely $R$ could be larger than the length of a segment, and We set $K$ and $R$ to appropriate values to ensure that multiple tasks can cover the entire region between $T^-$ and $T^+$.

Table 5: Hyper-parameters

| Dataset | IPC | $\beta$ | $R$ | $K$ | N | M | $T^-$ | $T^+$ | Synthetic Batch Size | Learning Rate (Label) | Learning Rate (Pixel) |
|---|---|---|---|---|---|---|---|---|---|---|---|
| CIFAR-10 | 1 | 0 | 2 | 3 | 80 | 2 | 0 | 4 | 10 | 5 | 100 |
|  | 10 | 0.2 | 4 | 4 | 80 | 2 | 0 | 20 | 100 | 2 | 100 |
|  | 50 | 0.2 | 8 | 4 | 80 | 2 | 0 | 40 | 500 | 2 | 1000 |
|  | 500 | 0.3 | 6 | 4 | 80 | 2 | 40 | 60 | 1000 | 10 | 50 |
|  | 1000 | 0.3 | 6 | 4 | 80 | 2 | 40 | 60 | 1000 | 10 | 50 |
| CIFAR-100 | 1 | 0.2 | 5 | 5 | 40 | 3 | 0 | 30 | 100 | 10 | 1000 |
|  | 10 | 0.2 | 10 | 4 | 80 | 2 | 0 | 50 | 1000 | 10 | 1000 |
|  | 50 | 0.2 | 12 | 4 | 80 | 2 | 20 | 70 | 1000 | 10 | 1000 |
|  | 100 | 0.2 | 12 | 4 | 80 | 2 | 30 | 70 | 1000 | 10 | 50 |
| Tiny | 1 | 0.3 | 7 | 3 | 60 | 2 | 0 | 20 | 200 | 10 | 10000 |
|  | 10 | 0.3 | 12 | 4 | 60 | 2 | 10 | 50 | 250 | 10 | 100 |
|  | 50 | 0.3 | 8 | 4 | 80 | 2 | 40 | 70 | 250 | 10 | 100 |

# E    ADDITIONAL EXPERIMENTS

## E.1    STABILITY OF TRAJECTORY MATCHING

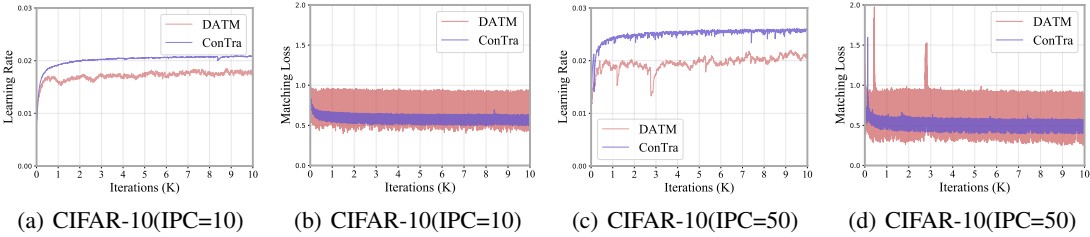

(a) CIFAR-10(IPC=10)        (b) CIFAR-10(IPC=10)        (c) CIFAR-10(IPC=50)        (d) CIFAR-10(IPC=50)

Figure 5: The learnable learning rate $\alpha$ and the matching loss during training on CIFAR-10, with IPC=10 and 50. The curve of our methods is much more smoothed

The negative correlations in matching different segments introduce another disadvantage to previous sampling-based TM methods, making the training highly unstable. Specifically, the matching loss frequently oscillates and cannot reduce to a relatively low level. Similarly, the learnable learning rates for the synthetic dataset exhibit the same issue, struggling to converge to a stable value.

Since we also employ the soft label trick from DATM, the primary difference between our method and DATM lies in our adoption of concurrent training. We compare the learning curves of ConTra and DATM in terms of learnable learning rate $\alpha$ and matching loss in Figure 5. The learning rate curve for ConTra is generally smoother and converges gradually. Although both methods exhibit

oscillations in the loss curves, the amplitude of oscillations for ConTra is significantly smaller than that of DATM, and the loss for ConTra is noticeably lower than DATM's loss. Without concurrent training, the negative correlation between different segments causes the loss of unsampled segments to increase. When these segments are sampled again, the loss rises to a higher value, resulting in substantial oscillations in the loss curve. Furthermore, the varying differences in matching loss across different segments may necessitate different learning rates, making it difficult for the learning rate to converge to a stable value.

## E.2 SCALABILITY

ConTra can scale up to ImageNet-1K by using TESLA (Cui et al., 2023). Specifically, TESLA only requires storing a single gradient computational graph even when unrolling $N$ steps updates of the synthetic dataset. We list the experimental results in Table 6, which demonstrate that concurrent learning can also yield improvements on large-scale datasets.

Table 6: Performance on ImageNet-1K (IPC=10)

| Method | TESLA | ConTra |
|---|---|---|
| Accuracy (IPC=10) | 17.8 | 20.4 |

## E.3 DONSTREAM TASK

The synthetic datasets generated via distillation are applicable not only to straightforward classification tasks but also to a range of downstream applications. For example, these datasets can function as proxies to accelerate model evaluation in Neural Architecture Search (NAS). Following (Zhao et al., 2021), we implement NAS on CIFAR-10 with the search space of 720 ConvNets varying in network depth, width, activation, normalization, and pooling. We try to identify the best network by training them for 100 epochs on the small synthetic dataset (IPC=10) for 100 epochs. For more details, please refer to (Zhao et al., 2021). The comparison with DC (Zhao et al., 2021) and Random is shown in Table 7. The two metrics used are the average test accuracy of the best-selected model and Spearman's rank correlation coefficient, which measures the agreement between the validation accuracy of the top 10 models trained on the proxy and the entire dataset. ConTra achieves higher accuracy and rank correlation than DC, indicating that it can reliably rank candidate architectures.

Table 7: NAS on CIFAR-10

| Method | Random | DC | ConTra | Whole Dataset |
|---|---|---|---|---|
| Accuracy(%) | 76.2 | 84.5 | 85.0 | 85.9 |
| Correlation | -0.21 | 0.79 | 0.83 | 1.00 |

## E.4 GENERALIZATION ACROSS VIT

Experiments in Section 6.3 verify that synthetic datasets exhibit good cross-architecture generalization across various CNN-based models. Another question worth exploring is whether similar results can be achieved under completely different architectures, *e.g.*, VITs. We train VITs on the synthetic datasets distilled by ConvNet. The test accuracy is listed in Table 8. We have two observations: (1) The performance is poor when the IPC is small. We speculate that this is because the VIT model is too large to achieve good results when training data is extremely limited; (2) On CIFAR-10, with IPC=1000, the performance improves significantly but is still far inferior to ConvNet. We hypothesize the reason is that the data distilled from gradient information based on ConvNet cannot be effectively applied to the different architecture of VITs.

## E.5 VISUALIZATION

We provide the visualization of Tiny Imagenet across different IPCs. In this part, our results are basically consistent with the visualizations in previous literature (Cazenavette et al., 2022; Zhang

Table 8: Cross-architecture generalization for VITs

| Method | VIT-Tiny | VIT-small | VIT-base | ConvNet |
|---|---|---|---|---|
| CIFAR-10 (IPC=1000) | 66.8 | 66.0 | 63.7 | 86.1 |
| CIFAR-100 (IPC=10) | 10.7 | 11.48 | 12.5 | 48.9 |

et al., 2023; Guo et al., 2023). When IPC is small, the synthetic dataset primarily consist of highly abstract images, representing the extraction of some class-wise generic easy patterns. As the IPC increases, the images gradually exhibit textures and details, enhancing their recognizability. A sufficient information capacity ensures that the synthetic dataset can retain patterns from both easy and hard samples.

## E.6 LIST OF SYMBOLS

Table 9: Cross-architecture generalization for VITs

| Symbol | Definition |
|---|---|
| $\mathcal{T}$ | Real dataset |
| $\mathcal{S}$ | Synthetic dataset |
| $C$ | Number of classess |
| $\tau^*$ | A complete expert trajectory |
| $\Theta_t^*$ | Parameters of the $t^{\text{th}}$ segment in the expert trajectory |
| $\theta_{t,0}^*$ | The starting parameters of $t^{\text{th}}$ segment in the expert trajectory |
| $\theta_{t,i}^*$ | The parameter obtained after $i$ optimization updates of $\theta_{t,0}^*$ |
| $\hat{\theta}_{t,0}$ | The starting parameters of $t^{\text{th}}$ segment in the student trajectory |
| $\hat{\theta}_{t,i}^*$ | The parameter obtained after $i$ optimization updates of $\hat{\theta}_{t,0}^*$ |
| $T$ | Number of segments in teacher trajectories |
| $M$ | The length of the expert trajectory |
| $N$ | The length of the student trajectory |
| $\mathcal{L}$ | Matching loss |
| $\mathcal{A}$ | A differentiable augmentation function in Eq. 1 |
| $\alpha$ | A learnable learning rate in Eq. 1 |
| $\epsilon_t$ | Accumulated error in the $t^{\text{th}}$ segment during evaluation |
| $\mathcal{I}_t$ | Initialization error in the $t^{\text{th}}$ segment during evaluation |
| $\delta_t$ | Matching error in the $t^{\text{th}}$ segment during evaluation |
| $\mathcal{U}_{\mathcal{S}}(f_\theta, N)$ | The updates of model $f$ after $N$ steps gradient decent on the synthetic dataset $\mathcal{S}$ |
| $\mathcal{U}_{\mathcal{T}}(f_\theta, N)$ | The updates of model $f$ after $N$ steps gradient decent on the real dataset $\mathcal{T}$ |
| $R$ | The distance between each segment that are simultaneously matched |
| $K$ | Number of tasks |

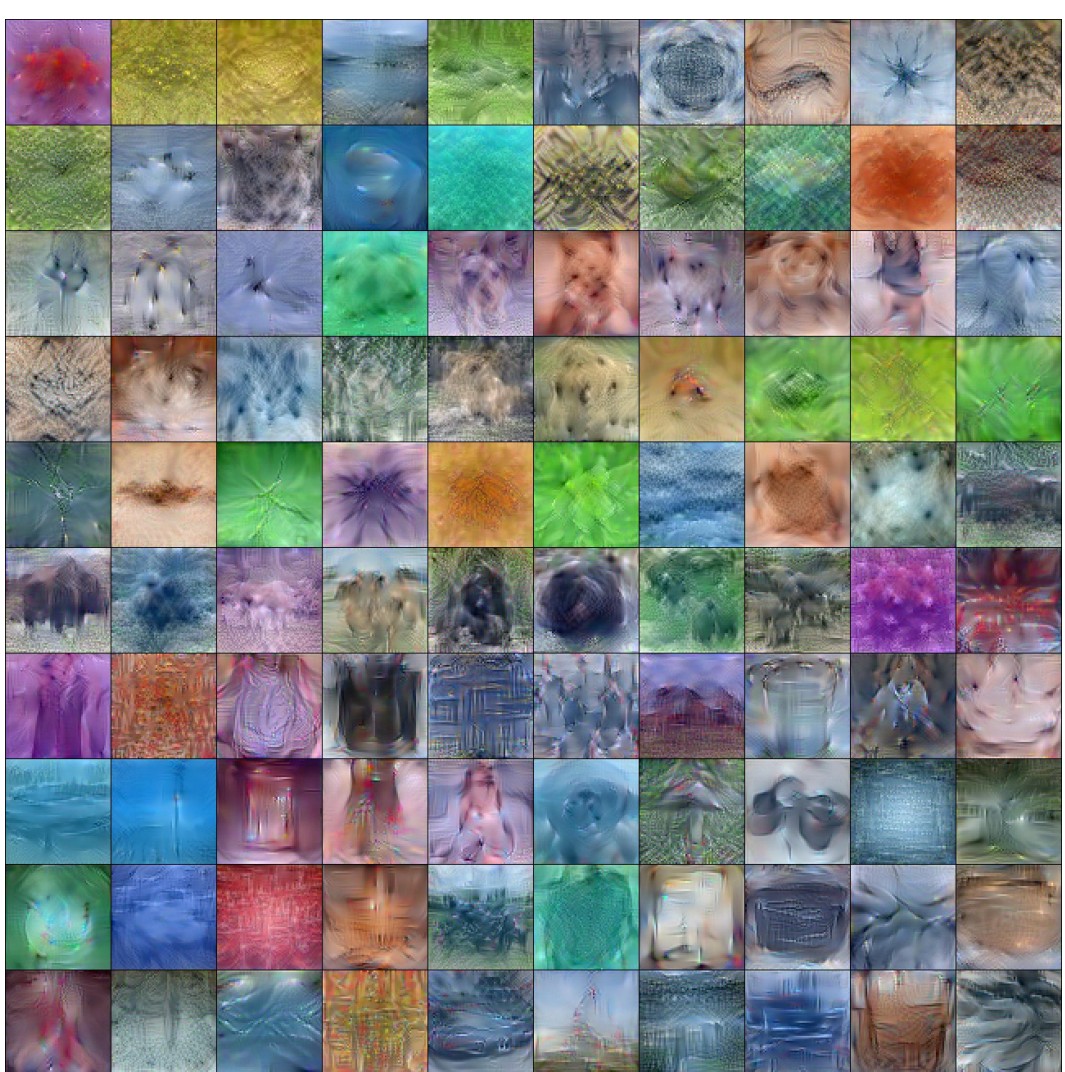

Figure 6: Tiny ImageNet (IPC=1): The visualization of the synthetic dataset (1/2).

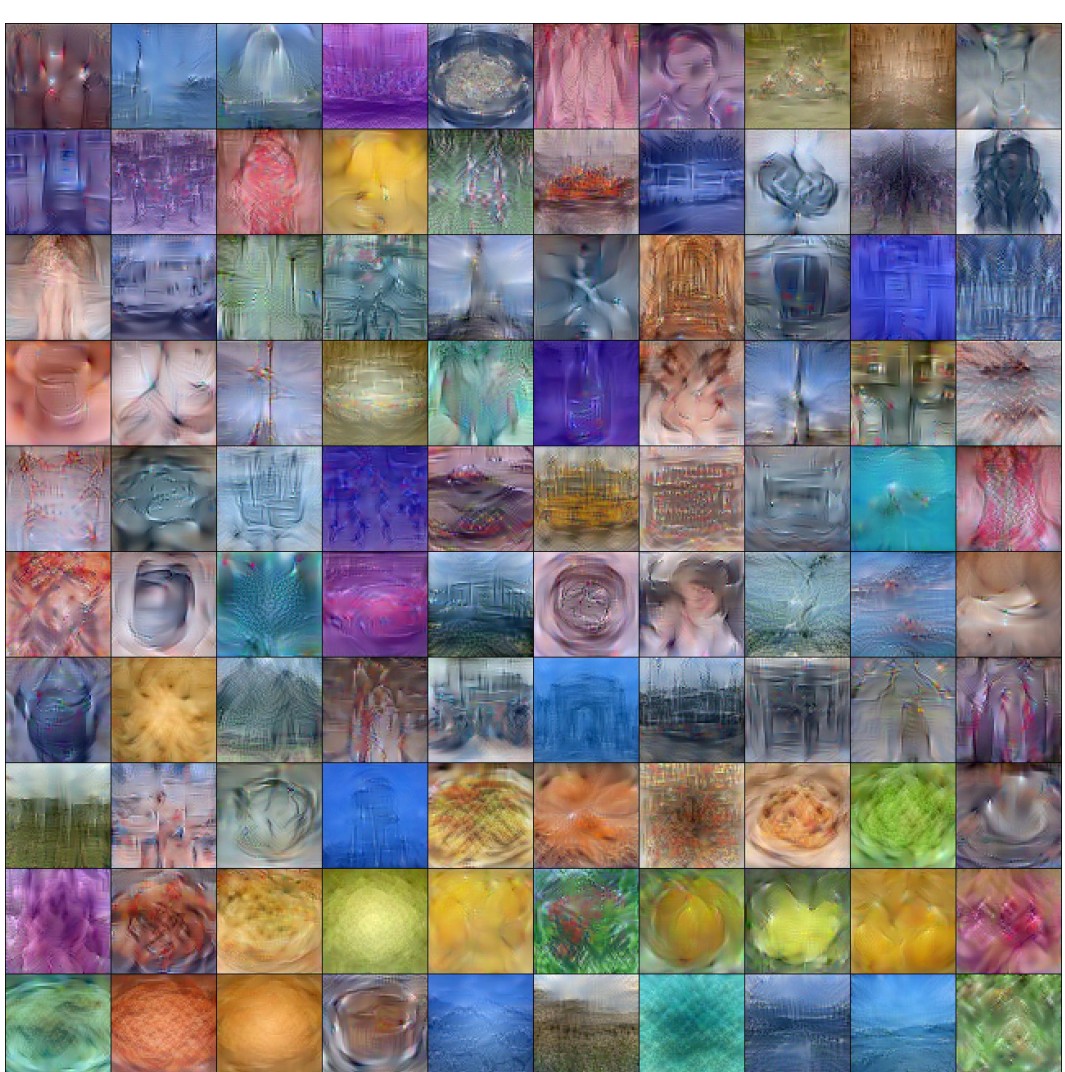

Figure 7: Tiny ImageNet (IPC=1): The visualization of the synthetic dataset (2/2).

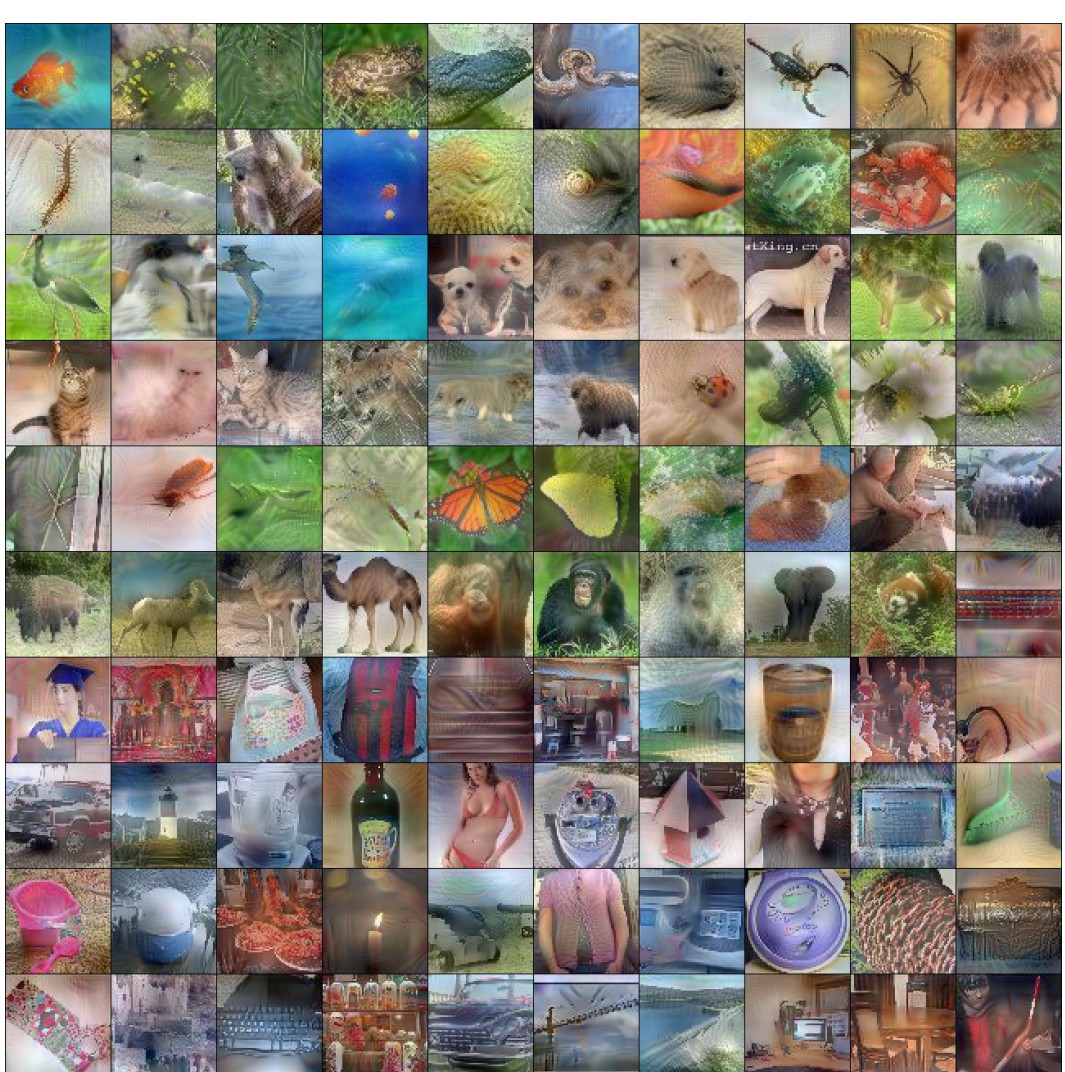

Figure 8: Tiny ImageNet (IPC=10): The visualization of the synthetic dataset (1/2).

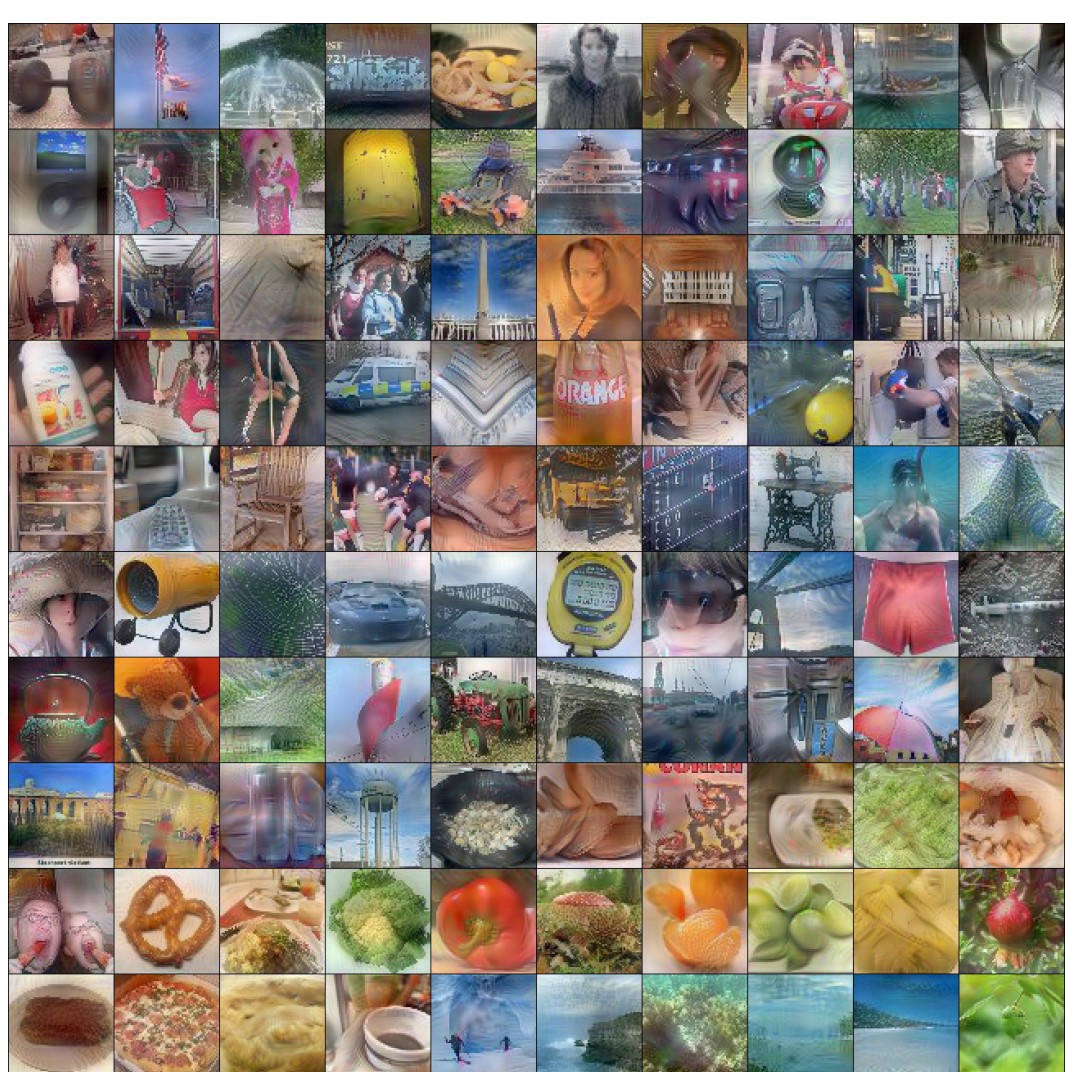

Figure 9: Tiny ImageNet (IPC=10): The visualization of the synthetic dataset (2/2).

1242
1243
1244
1245
1246
1247
1248
1249
1250
1251
1252
1253
1254
1255
1256
1257
1258
1259
1260
1261
1262
1263
1264
1265
1266
1267
1268
1269
1270
1271
1272
1273
1274
1275
1276
1277
1278
1279
1280
1281
1282
1283
1284
1285
1286
1287
1288
1289
1290
1291
1292
1293
1294
1295

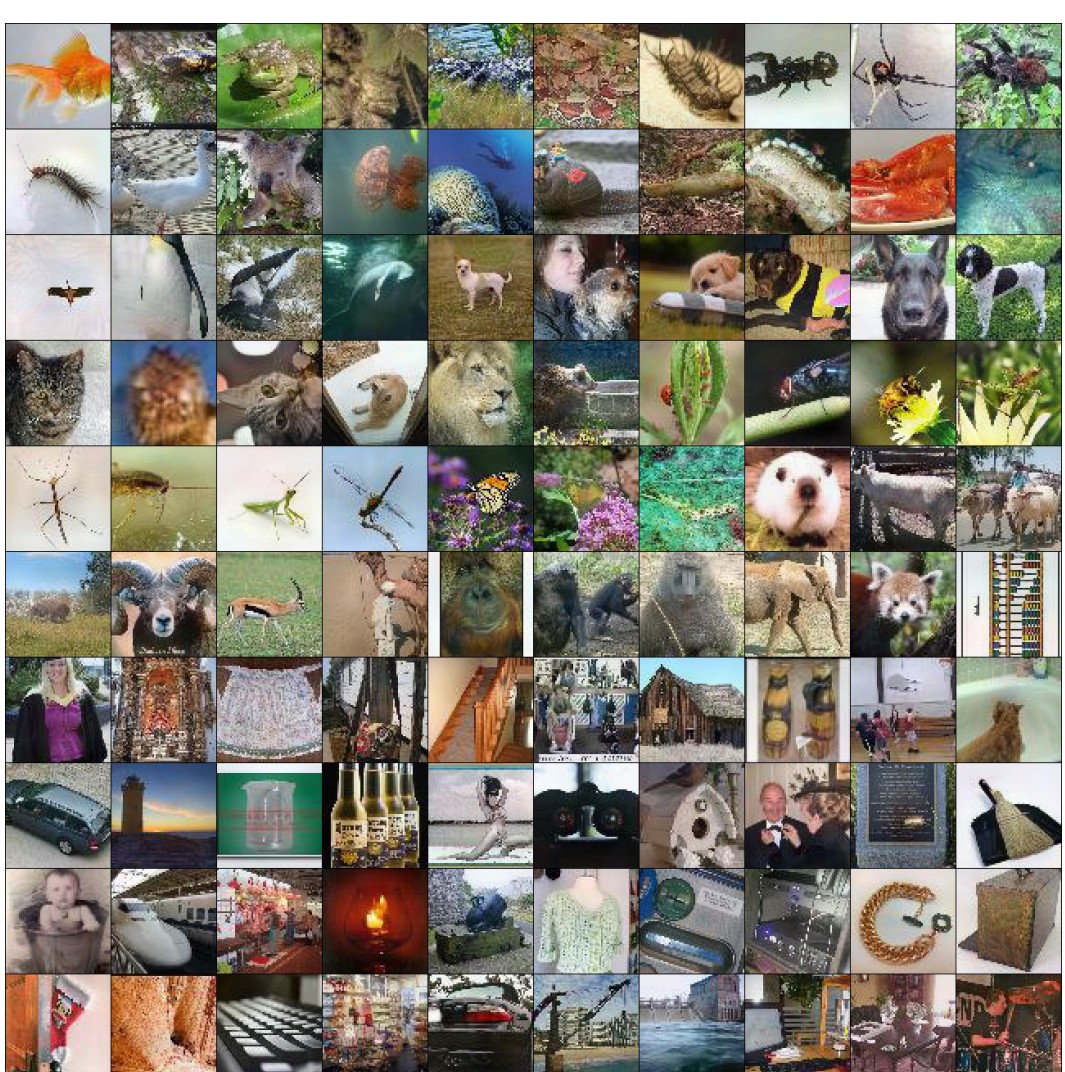

Figure 10: Tiny ImageNet (IPC=50): The visualization of the synthetic dataset (1/2).

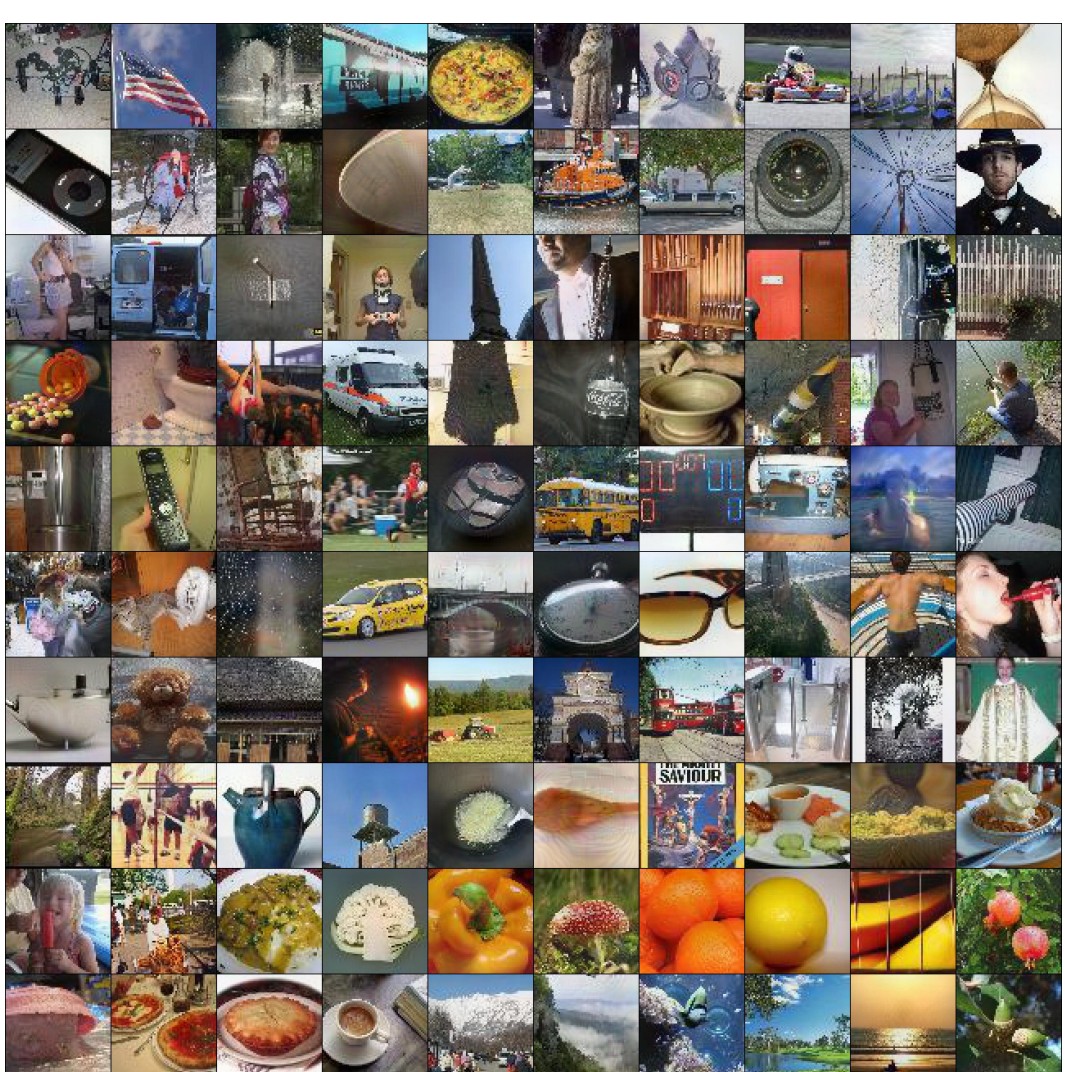

Figure 11: Tiny ImageNet (IPC=50): The visualization of the synthetic dataset (2/2).

