# OpenReview forum: "Enhancing Dataset Distillation with Concurrent Learning: Addressing Negative Correlations and Catastrophic Forgetting in Trajectory Matching"
_ICLR.cc/2025/Conference — Submitted to ICLR 2025_

### Official Review · Reviewer_kWAo · 2024-10-26

**Soundness:** 2
**Presentation:** 3
**Contribution:** 2
**Rating:** 5
**Confidence:** 3

**Summary:**

This paper tackles the problem that matching between different segments of the training trajectory may be negatively correlated. Existing methods use Trajectory Matching (TM), which optimizes the synthetic dataset by matching its training trajectory with the real dataset. However, they overlook the negative correlation between different trajectory segments, leading to performance degradation. The authors propose a concurrent learning-based TM method that matches multiple segments simultaneously, reducing errors. With exhaustive experiments, their approach outperforms previous methods across several benchmark datasets.

**Strengths:**

[1] The writing of paper is good.

[2] The analysis of the negative correlation on accumulated trajectory error is comprehensive. For example, the author clearly specifies the accumulated error to initialization error and matching error. Afterwards, the authors calculate the correlation to validate the phenomenon.

[3] The proposed cocurrent training methods is reasonable. The experimental results validate the effectiveness of the proposed methods.

**Weaknesses:**

[1] The description of the accumulated trajectory error is not intuitive. The notations defined in Sec. 4.1 are complex and there is no graphic illustration of these notations.

[2] The explanation of the negative correlation is not clear, especially when the IPC is low. In addition, the roles of training dynamics are not validated with experimental evidence.

[3] The novelty of the concurrent training to tackle the problem is rather limited. While reasonable, the author only simply leverages the multitask learning to tackle the problem. Since the improvement compared to previous state-of-the-art methods is not very significant, the contribution is not above the acceptance bar of ICLR.

**Questions:**

Please see the weakness.

---

> ### Author Response · Authors · 2024-11-14
> **First reply to reviewer kWAo**
>
> We appreciate the reviewer's constructive feedback, which helps enhance the clarity of our paper! Let us answer your question below.
>
> >**The description of the accumulated trajectory error is not intuitive.**
>
> Accumulated trajectory error represents the distance between the model optimized on the synthetic dataset and the model trained on the real dataset, and it is our ultimate optimization objective. It can be decomposed into two parts: initialization error and the matching error across all segments, and trajectory matching aims at decreasing the matching error. However, previous TM-based methods only optimized the error of a single segment at a time. Reducing the error of a single segment can be seen as one task, and therefore, our goal of reducing the overall matching error can be viewed as a multi-task, multi-objective optimization problem. When there are negative correlations among the objectives (tasks), optimizing only one of them will prevent the overall optimization goal from being effectively reduced.
>
> In the Appendix E.6 of the rebuttal revision, we add a table to show the definition of notations.
>
> >**The explanation of the negative correlation is not clear, especially when the IPC is low**
>
> In this work, negative correlation refers to the phenomenon where reducing the matching error of one segment causes the matching error of other parts to increase. At low IPC, the synthetic dataset can contain less information, making this issue more pronounced. Intuitively, we can think of the synthetic dataset as a hard drive, where IPC determines the amount of information that can be stored. When IPC is very low, if there are no constraints, the hard drive, in order to store new content (the trajectory matching of a new segment), will simply forget previously stored information (the segments that have already been matched).
>
> >**The novelty of the concurrent training to tackle the problem is rather limited, and improvement compared to previous state-of-the-art methods is not very significant**
>
> **[novelty of the proposed method]** As mentioned in Lines 347-348, we explored several other techniques. These solutions are clearly more technically novel and complex. However, we ultimately chose the simplest approach—concurrent training. On one hand, concurrent training is simple and effective, and it can be combined with any TM-based methods (Table 3) to yield improvements. We believe that the simpler the method, the stronger the underlying idea’s generality and novelty.
>
> On the other hand, another core novelty of this work is the theoretical analysis and empirical validation of the existence of negative correlations in trajectory matching, and our method is directly derived from this observation. We believe the discovery of negative correlation can offer valuable insights to the dataset distillation community.
>
> **[improvements compared with SOTA]** ConTra, as a plug-in method, can improve all trajectory matching methods. Compared to another plug-in method, PDD [4], the improvements we achieve are more substantial. For example, when the IPC is 10 on CIFAR-10, the improvement brought by PDD to MTT is 1.5%, whereas ConTra achieves an improvement of 2.5%. Moreover, DATM has already achieved lossless condensation at high IPC, and building upon this, but we still bring further improvements, which is quite challenging. And at small IPC, our improvement is highly significant. When IPC = 1, DATM shows a 0.7% improvement compared to the basic MTT, but we achieved a 3.1% improvement over the SOTA.
>
> If you have any other questions, we are happy to have further discussions.
>
> ## References
>
> [1] Chen, Xuxi, et al. "Data Distillation Can Be Like Vodka: Distilling More Times For Better Quality." The Twelfth International Conference on Learning Representations.

---

> ### Author Response · Authors · 2024-11-24
>
> Dear Reviewer kWAo,
>
> Thank you again for your questions and constructive review. We understand that your schedule is very busy, but the discussion period is ending in less than two days.
>
> We would greatly appreciate it if you could take a look at our reply to your review and let us know if you have any remaining questions. We look forward to addressing any remaining concerns before the end of the discussion period.

---

> ### Author Response · Authors · 2024-12-03
> **A warm reminder**
>
> Dear Reviewer kWAo,
>
> Today is the final day of the discussion phase, and we sincerely hope you could take a moment to review our reply. We would greatly appreciate it if you could let us know whether our responses have addressed your concerns.
>
> Best,
> authors of 5068

---

### Official Review · Reviewer_1dFc · 2024-11-01

**Soundness:** 3
**Presentation:** 3
**Contribution:** 3
**Rating:** 6
**Confidence:** 4

**Summary:**

The paper analyzes the challenges of dataset distillation, i.e., negative correlation between different segments of a trajectory to match and catastrophic forgetting problem. To address this, the authors formulate trajectory matching as a continual learning problem and propose a method called Concurrent Training-based Trajectory Matching (ConTra). It employs multi-task learning to simultaneously match multiple segments, rather than sequential learning used in previous works. Experimental results show that ConTra consistently outperforms existing trajectory matching methods on various datasets, thus demonstrating its ability to minimize accumulated matching errors and achieve lossless condensation.

**Strengths:**

1.	The analysis on negative correlation between different trajectory segments is detailed and solid, enriching the discourse on continual learning.
2.	The idea of utilizing concurrent learning to tackle negative correlation is simple but novel.
3.	Extensive experiments on multiple datasets and downstream tasks are quite convincing.

**Weaknesses:**

1.	The paper needs to be further polished. There are numorous typos, such as line 140 ‘a expert’->’an expert’, line 457 ‘s the range’->’so the range’.
2.	It would be more intuitive if there is a figure to show the differences/advantages of your proposed ConTra compared to the previous TM methods.
3.	The paper does not test the proposed method using different distillation and evaluation model size.

**Questions:**

1.	Is the proposed method sensitive to the model size used for distillation and evaluation?
2.	Does the distillation training time decease when using concurrent learning compared to sequential learning?

---

> ### Author Response · Authors · 2024-11-14
> **First reply to reviewer 1dFc**
>
> We would like to Thank you for your constructive review, as well as your positive feedback! Let us answer your question below.
>
> >**Typos**
>
> Thanks for pointing this out. We carefully check the draft and correct these typos in the rebuttal revision.
>
> >**It would be more intuitive if there is a figure to show the differences/advantages of your proposed ConTra compared to the previous TM methods.**
>
> We did not include a figure to show ConTra in the main text due to page limitations. The core difference between our approach and previous TM-based methods can be summarized in one sentence: previous methods sample and match one segment at a time, while we match multiple segments simultaneously to mitigate the issue of negative correlation
>
> >**The paper does not test the proposed method using different distillation and evaluation model size.**
>
> Thanks for this suggestion. In Table 2, we present the performance evaluated with three other convolutional networks of different sizes. In Table 8, we show the results for ViTs of varying sizes.
>
> To further explore the performance of ConTra using different distillation and evaluation model sizes, we conduct the following experiments (CIFAR-10, IPC=10). We observe that when IPC is small, distilling from ConvNet to ConvNet yields the best results, where D refers to distill, and E refers to evaluate. However, [1] found experimentally that when IPC = 1000, using a larger ResNet for distillation can achieve better performance. This may be because larger and more complex models contain more information, which requires a higher IPC to fully capture.
>
> |D\E|ConvNet|VGG11|ResNet18|
> |-|-|-|-|
> |ConvNet|68.3|50.6|52.9|
> |VGG11|47.3|42.7|40.6|
> |ResNet18|49.0|41.4|48.1|
>
> If you have any other questions, we are happy to have further discussions.
>
> ### References
> [1] Guo, Ziyao, et al. "Towards Lossless Dataset Distillation via Difficulty-Aligned Trajectory Matching." The Twelfth International Conference on Learning Representations.

---

> ### Author Response · Authors · 2024-11-24
> **Looking forward to your reply**
>
> Dear Reviewer 1dFc,
>
> Thank you again for your questions and constructive review. We understand that your schedule is very busy, but the discussion period is ending in less than two days.
>
> We would greatly appreciate it if you could take a look at our reply to your review and let us know if you have any remaining questions. We look forward to addressing any remaining concerns before the end of the discussion period.

---

> ### Author Response · Authors · 2024-12-03
> **A warm reminder**
>
> Dear Reviewer 1dFc,
>
> Today is the final day of the discussion phase, and we sincerely hope you could take a moment to review our reply. We would greatly appreciate it if you could let us know whether our responses have addressed your concerns.
>
> Best,
> authors of 5068

---

### Official Review · Reviewer_cujf · 2024-11-05

**Soundness:** 3
**Presentation:** 3
**Contribution:** 3
**Rating:** 5
**Confidence:** 4

**Summary:**

This paper addresses dataset distillation, aiming to create small synthetic datasets that enable models to achieve comparable performance to training on complete datasets. Due to convergence and stability challenges, Trajectory Matching methods typically match only segments of training trajectories, but they overlook negative correlations between different segments. The authors quantitatively analyze these correlations, finding that negative correlations can increase trajectory error and lead to catastrophic forgetting. To address this, they propose a concurrent learning-based TM approach that matches multiple segments simultaneously. Experiments show that this method outperforms previous approaches across various datasets.

**Strengths:**

1. This paper clearly identifies the problem, providing both theoretical and experimental analysis to demonstrate the existence of negative correlations between trajectory segments.

2. All the discussions in the paper are clear and straightforward.

3. The experiments contains all the necessary components with enough discussion

**Weaknesses:**

1. This paper includes a theoretical analysis of negative correlation; however, the theory presented in Section 4.1 primarily illustrates that training errors can accumulate across segments. While this is an important observation, it is not directly related to the negative correlation itself. A more detailed exploration of how these relate to negative correlation would strengthen the theoretical foundation of the proposed method.

2. The novelty of the proposed approach appears to be somewhat limited. To mitigate negative correlation, the method simply trains multiple segments concurrently, which is a strategy commonly employed as a baseline in continual learning scenarios.

3. The performance gains observed in the experiments are somewhat disappointing, particularly when considering the increased computational requirements associated with the proposed method. Given the additional complexity introduced, one would expect a more substantial improvement in performance to justify the costs involved (especially when compared to DATM). This raises questions about the effectiveness of the approach in real-world applications.

**Questions:**

See strengths and weaknesses.

---

> ### Author Response · Authors · 2024-11-14
> **First reply to reviewer cujf**
>
> We would like to thank the reviewer for the constructive comments! Let us answer the questions below.
>
> >**In Section 4.1,  A more detailed exploration of how these relate to negative correlation**
>
> By deriving the accumulated error, it becomes clear that trajectory matching aims to reduce the matching error of all segments. However, previous TM-based methods only optimized the error of a single segment at a time. Reducing the error of a single segment can be seen as one task, and therefore, our goal of reducing the overall matching error can be viewed as a multi-task, multi-objective optimization problem. When there are negative correlations among the objectives (tasks), optimizing only one of them will prevent the overall optimization goal from being effectively reduced. We do not further define assumptions and theoretically prove this point mainly because this conclusion is common, as similar results are found in continual learning [1], multi-objective optimization [2], and test-time training [3]. Section 4.1 theoretically demonstrates the multi-task nature of trajectory matching, while Section 4.2 empirically verifies the existence of negative correlations across multiple tasks, which leads to our proposed solution.
>
> >**The novelty of the proposed approach**
>
> As mentioned in Lines 347-348, we explored several other techniques. These solutions are clearly more technically novel and complex. However, we ultimately chose the simplest approach—concurrent training. On one hand, concurrent training is simple and effective, and it can be combined with any TM-based methods (Table 3) to yield improvements. We believe that the simpler the method, the stronger the underlying idea’s generality and novelty. On the other hand, another core novelty of this work is the analysis and empirical validation of the existence of negative correlations in trajectory matching, and our method is directly derived from this observation. We believe the discovery of negative correlation can offer valuable insights to the dataset distillation community.
>
> >**The performance gains observed in the experiments are somewhat disappointing**
>
> ConTra, as a plug-in method, can improve all trajectory matching methods. Compared to another plug-in method, PDD [4], the improvements we achieve are more substantial. For example, when the IPC is 10 on CIFAR-10, the improvement brought by PDD to MTT is 1.5%, whereas ConTra achieves an improvement of 2.5%. Moreover, DATM has already achieved lossless condensation at high IPC, and building upon this, but we still bring further improvements, which is quite challenging. And at small IPC, our improvement is highly significant. When IPC = 1, DATM shows a 0.7% improvement compared to the basic MTT, but we achieved a 3.1% improvement over the SOTA.
>
> Compared to DATM, a single iteration of ConTra is slower, but due to its faster convergence (according to Table 2) and more stable training (Figure 5), it does not incur much additional cost in practice.
>
> If you have any other questions, we are happy to have further discussions.
>
> ## References
>
> [1] Kirkpatrick, James, et al. "Overcoming catastrophic forgetting in neural networks." Proceedings of the national academy of sciences 114.13 (2017): 3521-3526.
>
> [2] Crawshaw, Michael. "Multi-task learning with deep neural networks: A survey." arXiv preprint arXiv:2009.09796 (2020).
>
> [3] Sun, Yu, et al. "Test-time training with self-supervision for generalization under distribution shifts." International conference on machine learning. PMLR, 2020.
>
> [4] Chen, Xuxi, et al. "Data Distillation Can Be Like Vodka: Distilling More Times For Better Quality." The Twelfth International Conference on Learning Representations.

---

> ### Author Response · Authors · 2024-11-24
> **Looking forward to your reply**
>
> Dear Reviewer cujf,
>
> Thank you again for your questions and constructive review. We understand that your schedule is very busy, but the discussion period is ending in less than two days.
>
> We would greatly appreciate it if you could take a look at our reply to your review and let us know if you have any remaining questions. We look forward to addressing any remaining concerns before the end of the discussion period.

---

> > ### Comment · Reviewer_kWAo · 2024-11-25
> >
> > Thanks for your comments. Although I highly appreciate your efforts, I still think the contribution is a little below the bar of ICLR. Therefore, I would like to keep my rating as boardline reject.  I also consider that the performance improvements can not fully empirically validate the effectiveness of the method.

---

> ### Author Response · Authors · 2024-11-27
>
> Thank you again for your feedback. Regarding whether we have empirically validated the effectiveness of our method, we would like to engage in further discussion with you on this point.
>
>
> ## **Overall Performance**
>
> In Table 1, we compare our method with state-of-the-art TM-based approaches. Our method consistently outperforms others across all datasets and IPC settings (1.18% on average), with particularly significant improvements in low-IPC scenarios (3.1%).  Compared to previous TM-based methods, the improvements achieved by our approach are non-trivial and notably significant.
>
> ## **Compared with another ICLR paper**
>
> In Table 3, we demonstrate that CT as a plug-in module, enhances the performance of any TM-based method. Compared to PDD [1], an ICLR 2024 accepted method that can also combine with various TM approaches, our improvements are even more pronounced.
>
> ## **Downstream task and large-scale dataset**
>
> Furthermore, for downstream tasks, we used NAS to showcase the superiority and benefits of our method. On large-scale datasets, our method achieves a significant improvement over TESLA [2], with performance increasing from 17.8 to 20.4.
>
> **We sincerely appreciate your valuable feedback and your time. Based on the points above, we humbly and earnestly hope that you may reconsider the contributions and experimental results of our work.**
>
> [1] Chen, Xuxi, et al. "Data Distillation Can Be Like Vodka: Distilling More Times For Better Quality." The Twelfth International Conference on Learning Representations.
>
> [2] Cui, Justin, et al. "Scaling up dataset distillation to imagenet-1k with constant memory." International Conference on Machine Learning. PMLR, 2023.

---

### Meta-Review · Area_Chair_j5tt · 2024-12-18

**Metareview:**

(a) Scientific Claims and Findings

The paper addresses the challenge of dataset distillation by proposing a Concurrent Training-based Trajectory Matching (ConTra) method. This approach aims to mitigate negative correlations between different segments of training trajectories, which can lead to increased trajectory error and catastrophic forgetting. By framing trajectory matching as a continual learning problem, ConTra employs multi-task learning to match multiple segments simultaneously. Reviewers note that the method outperforms previous approaches across various datasets, demonstrating its potential to minimize accumulated matching errors.

(b) Strengths

Reviewer cujf highlights the paper's clear identification of the problem and its theoretical and experimental analysis of negative correlations. 1dFc appreciates the detailed analysis and simple idea of using concurrent learning. kWAo commends the comprehensive analysis of negative correlation and the reasonable approach of concurrent training, with experimental results validating its effectiveness. Overall, the paper is well-written and provides extensive empirical evidence supporting the proposed method.

(c) Weaknesses

The reviewers point out several weaknesses. cujf notes that the theoretical analysis does not directly relate to negative correlation and that the novelty of the approach is limited, with performance gains not justifying the increased computational requirements. 1dFc mentions the need for further polishing of the paper and suggests including a figure to illustrate the advantages of ConTra. kWAo finds the description of accumulated trajectory error complex and the novelty of the concurrent training approach limited, with improvements not significant enough to surpass the acceptance bar for ICLR.

(d) Decision Reasons

On balance, AC agrees with negative points raised by the reviewers which outweigh slightly positive assessment of reviewer 1dFc, who did not engage in the discussion, nor provided a rationale to champion the paper for acceptance. The decision to reject the paper is based on the limited novelty and the modest performance improvements over existing methods, as highlighted by reviewers cujf and kWAo. While the paper provides a solid analysis and a reasonable approach, the contributions are not deemed significant enough to meet the standards of ICLR. Additionally, the increased computational complexity without substantial gains raises concerns about the practical applicability of the method. Despite the strengths in analysis and presentation, the weaknesses in novelty and impact lead to the decision to reject.

**Additional Comments On Reviewer Discussion:**

During the rebuttal period, the authors attempted to address the concerns raised by the reviewers, but the responses did not lead to significant changes in their evaluations.

Reviewer kWAo acknowledged the authors' efforts in addressing the feedback but maintained their position that the contribution of the paper is slightly below the ICLR acceptance threshold. They reiterated that the performance improvements presented in the paper do not sufficiently validate the method's effectiveness, leading them to keep their rating as a borderline reject.

Reviewer 1dFc, who was initially the only positive reviewer, did not participate in the discussion or provide further rationale to advocate for the paper's acceptance.

Reviewer cujf reviewed the authors' response but remained unconvinced about the paper's novelty, which they deemed insufficient for ICLR. Consequently, they decided to maintain their current rating, indicating a marginally below acceptance threshold.

In weighing these points for the final decision, the primary considerations were the consistent concerns about the paper's limited novelty and the modest performance improvements, as highlighted by reviewers kWAo and cujf. The lack of a strong advocate for the paper, particularly from reviewer 1dFc, further contributed to the decision to reject. Despite the authors' efforts during the rebuttal, the paper did not demonstrate the level of innovation and impact required for acceptance at ICLR.

---

### Decision · Program_Chairs · 2025-01-22

Reject